# Position: Web Agents Should Use Typed Actions Instead of Click-Based Browsing

**Linxi Jiang** [1]  **Rui Xi** [2]  **Zhijie Liu** [1]  **Shuo Chen** [3]  **Zhiqiang Lin** [1]  **Suman Nath** [3]

## Abstract

This position paper argues that building a reliable agentic Web requires shifting from low-level interaction primitives to typed actions supported by a semantic layer. Today's web agents primarily operate through clicks, keystrokes, and DOM manipulation, which leads to brittle long-horizon behavior, high execution cost, and limited auditability. We propose *web verbs* as a concrete design for this layer. A verb exposes a web operation as a typed function with structured inputs, structured outputs, and documented behavior, whether it is backed by a server-side Web API or a maintained client-side workflow. Verb calls can carry preconditions, postconditions, policy tags, and logging hooks, allowing agents to synthesize concise programs with explicit control flow and data flow and to produce checkable execution traces. Using representative case studies, we illustrate how verb-level composition can produce correct, reproducible outcomes, while browser agents using low-level interaction primitives may produce brittle behavior or incorrect reasoning. We conclude with a call to action on standardization, developer tooling, and community processes needed to make this semantic layer deployable and trustworthy at web scale.

## 1. Introduction

The Web is rapidly shifting from a place where people *browse* to a platform where software *acts* (Ning et al., 2025). With modern large language models (LLMs), it is increasingly realistic for users to delegate open-ended tasks through natural language (Liu et al., 2024; Plaat et al., 2025). A *web*

[1] The Ohio State University [2] University of British Columbia [3] Microsoft Research. Correspondence to: Shuo Chen, Suman Nath <{shuochen, sumann}@microsoft.com>, Zhiqiang Lin <zlin@cse.ohio-state.edu>.

*Proceedings of the 43rd International Conference on Machine Learning*, Seoul, South Korea. PMLR 306, 2026. Copyright 2026 by the author(s).

*agent* applies this idea to the Web: it interprets a natural-language task and acts across websites to complete goals such as booking trips, making purchases, or submitting applications (Nakano et al., 2021; Yao et al., 2022). This vision of an *agentic web* has become a central direction in recent agent research (Qin et al., 2025; Agashe et al., 2025a;b).

At the same time, practical web automation in the wild remains brittle, slow, and difficult to audit, especially for long-horizon and cross-site workflows (Kara et al., 2025). A key reason is that most agents still interact with the web through *low-level interaction primitives*, namely GUI actions such as clicks and keystrokes, as well as DOM manipulation (Prabhu et al., 2025; Lai et al., 2024). This interface forces agents to construct long and fragile traces, perceive and plan again after each small step, and produce executions whose intent and correctness are hard to check or reproduce. As a result, improvements in model capability do not consistently lead to reliable, efficient, or verifiable web automation in real deployments.

**We argue that stronger models alone are insufficient for a reliable agentic web without a semantic layer that exposes typed actions for common web tasks.** A typed action represents a web operation through input types, structured outputs, and expected behavior. Much current work on web agents has treated web interaction as a problem of learning better policies over the same low-level interaction primitives (Yang et al., 2025a). This framing has a structural limit. Without typed actions with stable semantics, agents will continue to spend much of their effort rediscovering brittle, site-specific click sequences and UI cases rather than composing correct and checkable workflows.

Our position does not replace progress in model capabilities, but argues that these capabilities need a better interface to the Web. It also parallels recent efforts to make web content more agent-friendly (Lù et al., 2025). For example, NLWeb (Microsoft, 2025) wraps the raw web with a semantic layer for retrieving and grounding web content through sources such as RSS (Hammersley, 2003) and Schema.org (Schema.org Community Group, 2011). This can improve what an agent *knows* about the web content, but it does not define what an agent can *do* on the web. The

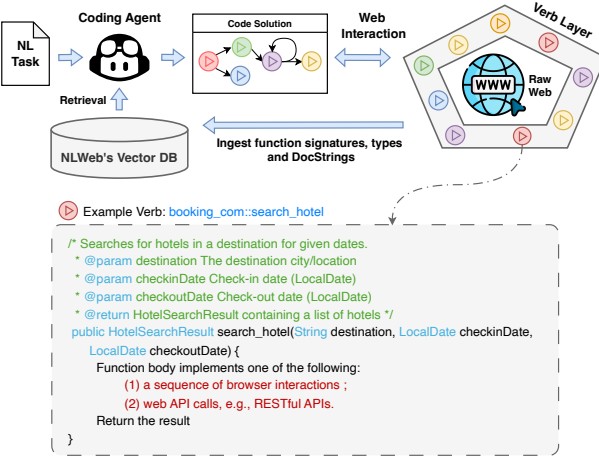

*Figure 1.* Overview of a verb-based workflow for typed actions on the Web. Websites expose common web operations as web verbs with typed inputs and structured outputs. Agents retrieve relevant verbs and compose them into executable procedures.

agentic web also needs stable interfaces for actions, such as searching for products or updating account settings.

We therefore recommend that **the web expose a semantic layer of typed actions that agents can invoke and compose.** We use *web verbs* as a concrete design for this idea. A verb is a typed, semantically documented function that exposes a web operation through a uniform interface. A verb can be implemented by calling a server-side Web API or by wrapping a robust client-side workflow using browser automation tools such as Playwright (Microsoft, 2020). In both cases, the agent interacts with the same typed verb interface and receives structured outputs for later computation. Figure 1 provides an overview of the proposed workflow, from publishing and indexing verbs to composing them into executable procedures.

A semantic layer of typed actions enables three properties that are difficult to obtain with low-level interaction primitives alone. First, it improves *reliability* by packaging stable semantics so that agents do not need to rediscover fragile internal steps on every run. Second, it improves *efficiency* by collapsing repeated perception and action cycles into a small number of reusable typed invocations. Third, it improves *verifiability* by making inputs and outputs explicit and checkable, supporting logging, auditing, and preconditions and postconditions at action boundaries. These properties are necessary for deploying web agents in settings where correctness and reproducibility matter.

In the rest of the paper, we first explain why both browser agents based on low-level interaction primitives and API agents based on server-side Web APIs fall short for end-to-end web tasks. We then introduce web verbs as a concrete form of typed actions and describe how agents

can compose them into short procedures with explicit control flow and data flow. Next, we present representative case studies that contrast low-level interaction traces with verb-level composition. We then summarize why the verb layer improves reliability, efficiency, and verifiability. Finally, we discuss alternative views, standardization needs, developer tooling, and open problems for making web verbs deployable at web scale.

**Code and Artifacts.** Our prototype, verb catalog, and task set are available at `https://github.com/nlweb-ai/MSR-Web-Verbs`.

**Conflict of Interest Disclosure.** The authors declare no financial conflicts of interest related to this work.

## 2. The Interaction Bottleneck in Web Agents

Most web agents today follow a perception-action loop. At each step, the agent observes the current state of the environment, selects a next move, executes an action, and then receives an updated state. This cycle repeats until the task is complete. In practice, the field has converged on two dominant interaction paradigms, distinguished by how actions are represented and executed.

- *Browser agents* act on the browser side, perceiving the DOM, screenshots, or both, and executing low-level interaction primitives, including GUI actions such as clicking, typing, scrolling, and navigation (Deng et al., 2023; Zhou et al., 2024; de Chezelles et al., 2025).

- *API agents* interact through server-side Web APIs, invoking operations over structured inputs and outputs whenever suitable APIs are available (Song et al., 2025; Lai et al., 2025; Yang et al., 2025b).

**Reliability is brittle under low-level primitives.** Low-level interaction primitives make reliable workflow synthesis difficult because the action abstraction is too fine-grained and carries little semantic information (Wang et al., 2025). Agents must compose many small operations whose meaning depends on page context. For example, a button click may correspond to "opening a drop-down date-picker for selecting the departure date on Alaska Airlines". This level of granularity is far below the semantics of natural task composition, such as "searching for Alaska Airlines flights with these criteria". Composing such fine-grained operations is especially problematic when workflows span multiple sites or require long chains of actions (Cheng et al., 2025). As the trace grows, small perception and action errors accumulate, making the task harder to complete correctly.

This brittleness is reflected in leading browser agents. IBM CUGA (Marreed et al., 2025), a top-performing system on the WebArena (Zhou et al., 2024) benchmark, achieves an average success rate of 0.617, yet performance varies

widely across sites. Its success rate on multi-site tasks drops to $0.354$, even though those tasks involve fewer records. This shows how cross-site composition stresses low-level interaction.

**Server-side Web APIs cover only part of the workflow.** API agents are limited not because server-side Web APIs are weak, but because server-side Web APIs usually expose only part of an end-to-end user workflow. Public Web APIs often support search, lookup, and retrieval, while the steps that complete a task remain tied to interactive web flows (Khan et al., 2021; Zhong & Su, 2013). A travel site may expose endpoints for flights and prices, yet seat selection, baggage add-ons, or refunds are handled through interactive web flows. Some of these steps may correspond to internal service calls, and they are not exposed as stable public APIs with an agent-facing contract.

Beyond public API coverage, many consequential workflows are bound to browser-session state (South et al., 2025). For example, canceling a subscription may require the user to be signed in, navigate a multi-step retention flow, and click a final confirmation button. Changing a shipping address may depend on cart state that only exists in the current browser session. Such operations are difficult to reproduce through a stateless public API call on behalf of an arbitrary user. Access may also require API keys, OAuth app registration, or strict quotas (Balash et al., 2022). These constraints make server-side Web APIs an incomplete interface for many end-to-end tasks that users would want agents to complete.

**Step-by-step interaction is costly and scales poorly.** With low-level interaction primitives, most progress comes from a long sequence of small interactions. Agents must repeatedly perceive the page and decide the next action. Each step triggers DOM or screenshot processing, LLM inference, and tool calls. The cost is significant even in routine tasks. Booking travel or completing checkout can require tens of interactions (Zhou et al., 2024; Xie et al., 2024). Cross-site workflows can easily exceed one hundred steps, increasing latency and token cost. This cost also repeats across users and task instances. Even when users ask agents to perform similar workflows, the agent must still reason through the next low-level action at each step rather than reuse a stable task-level operation.

These limitations suggest that neither browser agents based on low-level interaction primitives nor API agents based on server-side Web APIs provide a scalable foundation for reliable web agents. This motivates a semantic layer that exposes typed actions as the primary interface for composition, verifiability, and efficient execution, which we develop in the next section.

# 3. Verb Abstraction for the Agentic Web

This section gives a concrete design for the semantic layer introduced above. Our central recommendation is to expose common web operations as typed actions that agents can invoke and compose, rather than requiring agents to execute long sequences of low-level interaction primitives.

## 3.1. Web Verbs

As introduced in §1, *web verbs* are a concrete form of typed actions for the Web. A web verb is a high-level, typed, function-like interface for a web operation. Unlike low-level interaction primitives such as text entry and mouse clicks, a web verb specifies its inputs, outputs, and expected behavior. The goal is to make web operations stable units for planning and execution, rather than brittle traces over UI details.

Web verbs should satisfy several interface-level requirements.

- **Semantic clarity.** Each verb should state its intended operation explicitly in natural language so that its role is transparent to both agents and developers.

- **Typed parameters.** Inputs should have names and types, making it clear which arguments the agent must provide and how the verb can be used in workflows.

- **Structured results.** Outputs should be returned as structured objects or records so that subsequent steps can consume them directly without ad hoc parsing.

- **Composability.** Verbs should be easy to combine through data flow and control flow, allowing tasks to be represented as concise programs over meaningful operations rather than long traces of low-level interaction primitives.

- **Auditability.** Each invocation should expose inputs and outputs suitable for logging and checking, providing a basis for validation, debugging, and governance.

These requirements describe the minimal contract that makes a web verb easy to use, compose, and check.

> ***Our Recommendation:*** *Websites should expose high-value web operations as verbs with typed inputs, structured outputs, and clearly documented behavior.*

A collection of web verbs forms a site's semantic layer for agents, which we refer to as the *verb layer*. The verb layer should present a single typed verb interface, while allowing two implementation paths: server-side Web APIs and client-side workflows. When a site provides a suitable API, developers can wrap the endpoint as a verb, for example, Spotify's search functionality with parameters for query, item type, and market. When functionality resides in client-side workflows, developers can implement the verb using

browser automation tools such as Playwright (Microsoft, 2020). This use of automation is related to screen-scraping scripts and browser macros, but web verbs expose the workflow through an agent-facing typed verb interface rather than as a standalone script. In both cases, agents interact only with the typed verb interface. The distinction between server-side Web APIs and client-side workflows becomes an implementation detail rather than an agent-facing concern.

> **Our Recommendation:** *The verb layer should unify client-side workflows and server-side Web APIs behind a common typed verb interface that hides implementation details.*

This design also clarifies why web verbs are not merely public Web APIs under a new name. A public API usually exposes server-side functionality as a developer-facing product, with its own access model, documentation, versioning, quotas, and support obligations. A web verb instead exposes a web operation as an agent-facing typed action. It may be implemented by a server-side Web API, a client-side workflow, or a combination of both. This matters because many consequential workflows are not available as a single public API operation and may depend on browser-session state or on-page confirmation.

For browser-backed verbs, our claim is not that today's browser automation stack is already sufficient. Browser-backed verbs provide a practical path for existing sites while moving fragile interaction logic into a maintained implementation that can be tested, hardened, versioned, and reused. This shifts work from repeated runtime reasoning by each agent to shared offline engineering.

Figure 2 illustrates how a browser-backed verb can package a browser workflow. The example verb `get_direction` for Google Maps takes a source and destination as typed inputs and returns a structured result containing travel time, distance, and route. Internally, the implementation uses Playwright to (1) enter the source and destination, (2) select the driving option, and (3) read the travel time and distance from the rendered page. The key point is that this multi-step interaction is exposed as a single typed invocation, so agents plan and compose at the verb level rather than reasoning over each low-level interaction primitive.

## 3.2. Programmatic Composition over Verbs

With verbs as the abstraction layer, tasks are no longer carried out as step-by-step predictions of low-level interaction primitives. Instead, the agent generates explicit programs composed of verb calls. Because each verb has specified arguments and structured returns, the program can pass outputs into later calls instead of making the agent infer call details or parse ambiguous results in context. Within a pro-

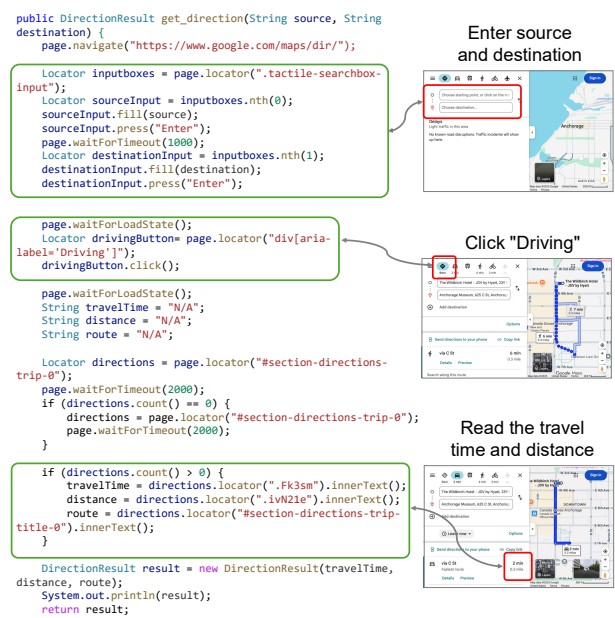

*Figure 2.* A browser-backed implementation of a web verb. The `get_direction` verb packages a browser workflow using Playwright, while exposing typed inputs and structured outputs to the agent.

gram, the results of a verb can be transformed, combined with other data, or used as conditions for subsequent steps. This makes the logic of the task explicit in code: the synthesized program specifies not only which verbs are invoked but also how their outputs are processed and connected.

The composition of verbs is not limited to a linear flow. Programs can include control structures such as conditionals, where the next operation depends on intermediate results, or loops, where the same verb is applied repeatedly over a collection of inputs. Branching and iteration are expressed directly in the program rather than decided one step at a time during execution. This stands in contrast to browser agents, which must repeatedly observe the current state and predict the next low-level action. By operating at the level of verbs, the agent can construct a procedure whose operations and dependencies are explicit. The role of the agent shifts from predicting primitive steps to composing structured programs that preserve the task logic.

> **Our Recommendation:** *Agents should plan and execute tasks by synthesizing short procedures that compose typed verb calls with explicit data flow, control flow, and boundary checks.*

Programmatic composition does not remove runtime adaptation. A well-designed verb can handle routine variability inside its implementation, such as retries, session checks, transient pop-ups, or minor UI differences. If an unexpected

situation falls outside the verb's handling logic, the verb can return a structured failure or recovery signal. The agent can then replan at the verb level, choose another verb, or fall back to low-level interaction primitives when needed. It is also worth noting that today's Web was designed primarily for human users, while an agent-friendly Web should provide more stable interaction surfaces for web agents, which we discuss in §7.

## 4. Case Studies

To illustrate the practical value of our recommendations, this section presents two representative web tasks selected from our prototype evaluation. In that evaluation, we implemented verbs for more than ten websites and tested over one hundred distinct complex tasks, all of which completed successfully. The full task set and verb artifacts are available in our project repository. We use the two tasks below to contrast verb-level composition with low-level interaction traces. Using web verbs, an agent can express the workflow as a compact procedure with explicit intermediate values, control flow, and task constraints. Under low-level interaction primitives, the same goal expands into long step-by-step traces.

To make the examples concrete, we describe a small verb layer with web verbs for a few widely used sites. It includes a Google Maps verb that returns travel time and distance between two locations and a set of Amazon verbs that return structured product attributes such as price and rating. These verbs expose typed inputs and structured outputs, while their implementations rely on available endpoints when possible and use browser automation when needed. Additional details and the verb catalog are provided in Appendix A. For reference, we also use browser-use (Müller & Žunič, 2024) to illustrate how a browser agent using low-level interaction primitives would approach the same tasks.

### 4.1. Travel Planning

This case study shows how a travel planning task can be expressed concisely with the verb layer, and why the same task is easy to mishandle with low-level interaction primitives. The user request is as follows:

> **User Task**
>
> *"I will travel to Anchorage for three days, and I would like to visit two museums each day. Please recommend good museums and hotels, and then rank the hotels by the total distance to all the selected museums."*

The task is not just to recommend museums and hotels. It also asks for a specific ranking: for each candidate hotel,

compute its distance to each selected museum and aggregate these distances into a single total. Figure 3 contrasts low-level interaction traces with verb-level composition.

With web verbs, the task is expressed as a short executable procedure with explicit intermediate results. It naturally decomposes into two steps. First, the agent identifies candidate museums and hotels in Anchorage by invoking the NLWeb (Microsoft, 2025) ask interface wrapped as a verb, which returns structured entities with metadata such as name, description, and location. Second, to rank hotels by total distance to the selected museums, the agent invokes the Google Maps verb get_direction, which returns travel time and distance as structured outputs. Figure 4a shows the agent-synthesized nested-loop procedure. Using get_direction, the synthesized program loops over hotels and museums, accumulates pairwise distances into a per-hotel total, and then sorts hotels by that total.

With low-level interaction primitives, the intended computation can be lost when the user request is translated into a long sequence of UI steps. In our comparison, a browser agent did not compute the distance from each hotel to each museum and sum the results. Instead, it chained the museums as waypoints in a single route and treated that route length as the total. The output looked plausible, but it answered a different question. The issue was not a failed click. It was a mismatch between the user-specified computation and the computation carried out through the UI.

### 4.2. Furniture Shopping

The second case study considers a user who has just moved into a new apartment and needs to purchase essential furniture, including a bed frame, mattress, desk, chair, floor lamp, air purifier, and carpet. The user request is as follows:

> **User Task**
>
> *"I just moved into a new home and need to buy furniture, including a bed frame, mattress, desk, chair, floor lamp, air purifier, and carpet. Please find three candidate products for each category, and then select a combination that maximizes overall rating while keeping the total cost within a budget of $1,000."*

This request combines product search with a constrained selection problem. The agent must choose one item per category, stay within a global budget, and maximize an objective over the entire set.

By invoking web verbs, the agent retrieves candidates for each category with structured fields such as price and rating, reducing the remaining step to a standard selection problem. The agent then synthesizes a program that searches over

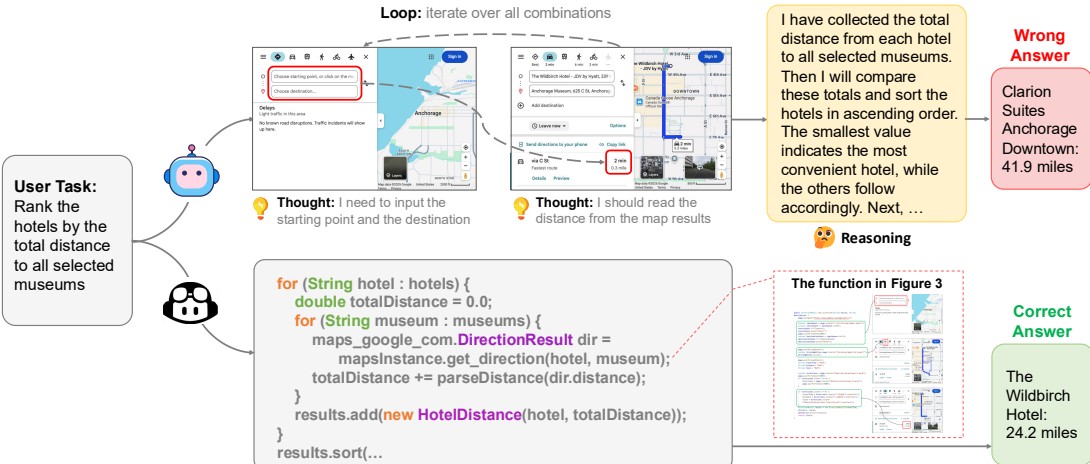

*Figure 3.* Workflow comparison between verb-level composition and low-level interaction traces. Web verbs allow agents to compose short executable procedures, while low-level traces expand into long sequences of actions.

candidate combinations, computes the total cost and total rating, and returns the highest-scoring configuration that stays within the $1,000 budget. Figure 4b shows the agent-generated loop that evaluates combinations and updates the best set as it searches.

With low-level interaction primitives, the budget constraint and the global objective are harder to enforce reliably across categories. Browser agents may fall back to local choices, selecting items one by one and checking the budget only after several selections have been made. In our comparison, the agent adopted a greedy strategy, selecting items until the budget was exceeded and then stopping. This left some categories unfilled and skipped lower-cost alternatives that would have satisfied the constraint. The final result does not implement the stated optimization objective under a global budget.

### 4.3. Lessons from Case Studies

Across both case studies, well-structured web verbs make the task logic explicit in a form that agents can follow and check. Because verbs return structured outputs, the agent can write short procedures with visible intermediate values, standard control structures such as loops and conditionals, and direct checks against the user's requirements. In the travel task, this means computing hotel-to-museum distances and summing them according to the ranking rule. In the shopping task, this means evaluating complete product sets under a single global budget.

With low-level interaction primitives, the same logic must be carried through a long sequence of actions where intermediate values are rarely explicit. The agent must repeatedly translate its plan into UI steps while tracking changing page

```
for (String hotel : hotels) {
    double totalDistance = 0.0;
    for (String museum : museums) {
        maps_google_com.DirectionResult dir =
            mapsInstance.get_direction(hotel, museum);
        totalDistance += parseDistance(dir.distance);
    }
    results.add(new HotelDistance(hotel,
↪    totalDistance));
}
```

*(a)* Hotel ranking

```
for (List<Item> combo : allCombinations(items)) {
    int cost = sumPrice(combo);
    int score = sumRating(combo);
    if (cost <= budget && score > bestScore) {
        bestScore = score;
        bestSet = combo;
    }
}
```

*(b)* Furniture selection

*Figure 4.* Synthesized program fragments for verb-level composition in the two case studies. (a) Ranking hotels by summing distances from each hotel to the selected museums. (b) Selecting one item per category under a global budget.

state, which makes it easier to drop constraints or change the computation without noticing. In the travel task, the browser agent chained museums as waypoints in a single route instead of summing hotel-to-museum distances. In the shopping task, it returned a result that did not satisfy the budget constraint over the full set. Long traces also create long interaction histories that must stay in context, making it harder to keep intermediate results and task constraints consistent across steps.

Overall, low-level interaction traces were more likely to diverge from the user-specified computation. By contrast,

verb-level composition followed the stated requirements more directly and produced results that matched the intended logic.

## 5. Why the Agentic Web Needs the Verb Layer

The case studies show the practical value of the verb layer and why it is needed beyond these examples. We summarize these reasons along three dimensions: reliability, efficiency, and verifiability.

**First, verbs improve reliability by providing stable semantics.** Each verb exposes a predefined operation with a maintained implementation, so the agent does not need to infer concrete low-level interactions from the task description and the current webpage structure. This leads to more stable behavior and reduces the variation that comes from step-by-step control. Low-level interaction primitives carry little semantic meaning because each click or keystroke is bound to layout coordinates, element state, or interface styling without conveying the intended operation independently. By contrast, verbs expose explicit semantics through typed inputs and structured outputs, decoupling the agent's reasoning from surface-level features such as page layouts or element identifiers. As a result, workflow logic can remain stable even when interface details change. For example, retrieving directions between a hotel and a museum through raw interactions would require many fragile steps, whereas the verb `get_direction` exposes the operation as a single typed call backed by one maintained implementation.

**Second, verbs improve efficiency through structured composition and reuse.** Executing a task with low-level interaction primitives often expands into a long trace of clicks and keystrokes, and even API agents often need custom code to bridge gaps. Verbs instead expose operations as high-level building blocks that can be composed into concise programs with explicit control flow and data flow. This reduces the number of perception and reasoning cycles, turning tasks that once required many interactions into a handful of stable calls. It also changes how automation cost is paid. With browser agents, every user invocation forces the agent to rediscover and re-execute a fragile interaction trace. With verbs, the cost of understanding, hardening, and maintaining a workflow is concentrated in one reusable implementation by the site developer, a third-party maintainer, or a community tool builder, and then amortized across many agents and user tasks. The result is faster execution, lower repeated inference, perception, and tool-call cost, and better reuse of knowledge across sites.

**Third, verbs support verifiable execution.** Neither low-level interaction traces nor ad hoc API sequences provide clear evidence of what an agent has done. By contrast, verb calls are explicit operations with typed inputs and struc-

tured outputs, so they can be logged, checked, and audited. This transparency enables correctness checks, debugging of failed executions, and enforcement of site-level policies such as authentication requirements or safety constraints. In practice, agent behavior becomes a sequence of interpretable action invocations rather than an opaque low-level trace.

Beyond these technical benefits, the verb layer supports a larger goal for the agentic web. **The goal is not only to build agents for today's web, but also to make the web easier for agents to use.** With web verbs, sites expose stable typed actions and agents compose them into workflows. This split of roles lets sites provide the operational interface, while agents provide task reasoning.

The relationship is similar to the role of programming languages. Typed actions relate to low-level interaction primitives in the same way that high-level languages relate to assembly. High-level languages let developers write structured programs without reasoning about each register and instruction. Likewise, web verbs let agents construct workflows without reasoning about each click and keystroke. The key change is the level of abstraction: the underlying operations remain precise, but agents reason over typed actions rather than fragile low-level traces.

## 6. Alternative Views

This section discusses several alternative views on our central claim.

**View 1: Stronger models make GUI-based browsing practical.** A common counterposition is that the browser already provides a universal interface through the GUI and DOM. Any site that a human can use could, in principle, be controlled by an agent through perception and low-level interaction primitives such as click, type, and scroll. From this view, the most direct path is to keep improving models, training regimes, and benchmarks for GUI grounding and long-horizon control, rather than adding a separate action layer (Xue et al., 2026; Zhou et al., 2025). However, the choice of interface still matters even when stronger models make the mapping from intent to low-level actions more accurate. Completing tasks through low-level interaction primitives often requires long, stateful traces, where the agent must carry forward and correctly interpret many intermediate UI states. As the trace grows, context management becomes harder, small errors accumulate, and the risk of semantic drift increases (Tian et al., 2025; Zhang et al., 2025). These failure modes are difficult to remove by training alone because they come from the length and structure of the interaction, not only from perception accuracy. Typed actions reduce this burden by changing the unit of interaction. They expose a web operation through typed inputs, structured outputs, and documented behavior. This lets the agent express

a workflow as a short procedure with explicit intermediate values, rather than as a long sequence of low-level steps. Stronger models still help in this setting, since they can choose the right typed actions, fill in arguments more accurately, and synthesize better procedures. The agent also relies less on keeping interaction details in context, which can lower token and compute cost.

**View 2: Typed actions will not cover enough of the Web.** A common concern is that typed actions can only cover a small fraction of real web tasks. The Web contains diverse sites, long-tail workflows, and user goals that may not match any predefined action. From this view, typed actions may help for frequent operations such as search, checkout, or account updates, but the open-ended nature of the Web would still force agents to rely on low-level browsing for many tasks.We agree that typed actions will not cover the entire Web at the beginning. Our position is that this is not a decisive objection to the verb layer. Many delegated tasks are long-tail at the natural-language level, but they often decompose into a smaller set of recurring web operations. Exposing these operations as typed actions would already reduce the length and fragility of many workflows. The first goal is therefore not complete coverage of every possible interaction, but reliable coverage of common and consequential operations where stable semantics matter most. Coverage should also be understood as something that can improve after an interface becomes widely used. This has happened before on the human-facing Web. Early web interfaces were often inconsistent, inefficient, and difficult to use. As more users depended on websites for everyday tasks, many sites invested heavily in design, performance, and accessibility because better UIs lead to higher task success, repeat use, and revenue (Kuan et al., 2005; 2008). If web agents become a common way for users to access services, a similar process can occur for web verbs. Web developers would have incentives to spend time implementing more reliable verbs for core workflows and expanding verb coverage to more site functionality.

**View 3: Web verbs will face the same adoption barriers as public APIs.** Another concern is that the same forces that limit public Web APIs may also limit web verbs. Developers may not have time to maintain an additional interface, sites may have limited incentives to support automated access, and some services may rely on human interaction for advertising, preference collection, or consent. These concerns identify real deployment questions for any agent-facing web interface. The difference is that web verbs change how adoption can begin. A public API usually requires a site to expose backend functionality as a stable external developer product. A verb can instead be layered on top of the existing web by wrapping the same client-side workflow that a user already performs. This lowers the re-architecture burden because the site need not redesign the boundary between frontend and backend before exposing an agent-facing action. The deployment path can also begin outside the site itself. Third-party developers and community tool builders can create useful verbs by packaging browser-backed workflows, similar to how browser extensions and community-maintained automation scripts are built around existing sites. These community verbs may be less authoritative than site-maintained verbs, but they provide an incremental path for experimentation, reuse, and demand discovery. A shared community ecosystem can also help verbs improve over time: popular verbs can be tested across users, repaired when sites change, and refined into more stable interfaces. If a community verb becomes widely used, the site may replace it with an official version that is more stable, better governed, and better matched to site policy.

Finally, web verbs need not eliminate the human-facing experience. A verb can implement full automation for low-risk tasks, but it can also implement handholding automation in which the agent prepares each step and the user confirms consequential transitions. This preserves opportunities for user consent, policy acknowledgement, advertising-supported experiences, and preference collection when those are part of the service model. In this sense, web verbs are not a proposal to bypass the existing web, but a proposal to expose recurring user actions through typed and checkable interfaces.

## 7. Standardization, Community, and Open Problems

**Standardization.** Today's Web was designed primarily for human users, with dynamic interfaces that can be useful for people but brittle for agents. An agent-friendly Web should provide more stable interaction surfaces for web agents. The verb layer sits between the raw web and the agent, standards are needed for how web verbs refer to HTML elements and how web verbs are exposed to agents. Without such conventions, the verb ecosystem could fragment into overlapping, incompatible, or poorly documented actions. Standardization is therefore a core requirement of the proposal, not an optional refinement. On the HTML side, verbs need a stable way to refer to the elements they operate on, since existing selectors often change during routine redesigns or refactors. A standard could add a new attribute, for example `stable-public-locator`, to mark elements that are part of a site's public interface. The locator could identify an element through a DOM path, similar to CSS selectors. Unlike ordinary class names, however, it would be a compatibility commitment. Its value should remain stable across site updates, since changes can break existing verbs. The standard could also recommend short, descriptive identifiers, which improve readability and make the mapping easier to inspect. On the verb side, we also need a standard for

verb registration. Verbs should live in a global namespace, with each web domain controlling its own names, similar to DNS. Each verb would have a fully qualified name, such as `amazon.com::addToCart`. The standard should also define how to specify signatures, types, and docstring descriptions, along with safety specifications such as preconditions, postconditions, and permission manifests.

**The developer community.** Developers will play an important role in building an agentic web. A similar whole-community effort occurred when mobile computing became an evident paradigm shift. The original web was designed for PCs, so it required web developers to deliberately "wrap" it in order to support mobile applications. The shift toward agent-friendly interaction represents a comparable change. Developers are well positioned to implement verbs because they know their sites' semantics, constraints, and expected changes. To make this practical at scale, developer tools are needed to reduce the effort of building and maintaining verbs. A simple but effective idea is a tool that records a browser action trace and turns it into a function that calls an automation library such as Playwright. In this sense, the same AI methods used for GUI-based browsing can help developers create reusable verbs offline, rather than asking every user-facing agent to rediscover the workflow at runtime. As we discussed in §6, community contributions can help the verb layer grow before every site provides an official implementation. Third-party developers and community tool builders can publish browser-backed verbs for useful workflows, share them across agents and users, and use this process to reveal which workflows have recurring demand. These community verbs may later guide official site-maintained implementations that are more stable, better governed, and better aligned with site policy.

**Other open problems.** Several open problems remain important for future research. We highlight four areas: verb granularity, retrieval, security, and benchmarks.

Verb granularity is a key design question because it directly affects which workflows agents can compose. If verbs are too coarse, they may hide intermediate choices that agents need for comparison, filtering, or cross-site composition. For example, a shopping site could expose search, add-to-cart, and checkout as separate verbs, or collapse the entire purchase flow into one coarse verb. The former gives the agent more flexibility, while the latter may make tasks such as comparing products before purchase difficult or impossible. If verbs are too fine-grained, the abstraction advantage is lost and agents again face long traces of small steps. The community will need design guidelines and benchmarks that favor reusable operations with meaningful intermediate values, similar to how GUI design conventions improved over time.

For retrieval, RAG is a natural starting point for selecting relevant verbs from a library. With a small verb database, a basic retrieval setup is often sufficient. When the number of verbs grows to web scale, retrieval must account for more than natural-language descriptions. It should also use verb signatures, input and output types, site context, and preconditions so that agents select verbs that are not only relevant, but also callable in the current workflow.

Security and permissions remain critical challenges for any web agent. The verb layer gives agents explicit programming abstractions for web interaction, so a solution can be expressed directly as code over typed actions. In this setting, many security and permission requirements can be stated as preconditions and postconditions, then checked statically or dynamically. This creates research questions about how to model real web policies, user consent, authentication requirements, and site-level safety constraints as logic conditions.

Finally, benchmarks will be essential for evaluating this direction. Composing verbs allows agents to attempt complex, end-to-end tasks, but existing benchmarks were not designed for this level of abstraction. New benchmarks should include multi-site scenarios with explicit task constraints, security requirements, and objective success criteria. They should test not only whether an agent reaches a final answer, but also whether the composed workflow respects user intent, site policies, and typed action boundaries.

## 8. Conclusion

We argue that web agents should operate on a semantic layer of typed actions, rather than defaulting to low-level interaction primitives for real-world web services. We use web verbs as a concrete design point. A verb exposes a web operation as a typed function with structured inputs, structured outputs, and documented behavior, whether it is backed by a server-side Web API or a packaged client-side workflow. Verb-level procedures make intent, control flow, and data flow explicit. They shorten execution traces, reduce reliance on fragile UI details, and provide natural boundaries for policy checks, auditing, and controlled execution. Moving toward typed actions requires more than a better agent model. It requires a web ecosystem that makes verbs easy to create, share, maintain, and trust. Developer tools should help turn existing web workflows into reusable typed actions, while community processes can help useful verbs emerge before every site provides an official implementation. Standards for signatures, safety metadata, and registration are needed so that verbs can be discovered, composed, and checked across sites. With this ecosystem in place, web agents can move beyond repeatedly reconstructing routine workflows from low-level interactions and instead use stable typed actions as the common language between sites and agents.

## Acknowledgements

The OSU authors were partially supported by NSF Award CNS-2112471 and a Safety Science Award from Schmidt Sciences.

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

# A. Details of Implemented Verbs

As introduced in §4, our prototype semantic layer implements a small set of verbs expressed as typed functions. This appendix provides the complete catalog of the verbs used in our case studies, limited to Amazon and Google Maps. Each entry includes a concise description of the operation, the expected input parameters, and the structure of the returned output.

For consistency and readability, each entry follows the same format: (1) a natural-language description of the operation, (2) the typed parameters required to invoke it, and (3) the structured form of the returned output.

*Table 1.* Detailed specifications of the web verbs

| Website | Verb | Description |
|---|---|---|
| Amazon | `clearCart` | Clears all items from the Amazon shopping cart. |
| | `addItemToCart` | Adds an item to the cart and returns cart information. 
 **@param item**: The item to search for and add to cart 
 **@return** CartResult containing cart items and their prices. |
| GoogleMaps | `get_direction` | Gets travel information between two locations. 
 **@param source**: The starting location 
 **@param destination**: The destination location 
 **@return** DirectionResult containing travel time, distance, and route. |
| | `get_nearestBusinesses` | Retrieves a list of nearby businesses based on a reference point and description. 
 **@param referencePoint**: The location from which to find nearby businesses (e.g., "Seattle, WA"). 
 **@param businessDescription**: The type of business to search for (e.g., "coffee shop"). 
 **@param maxCount**: The maximum number of businesses to return. 
 **@return** NearestBusinessesResult containing a list of the nearest businesses. |
| | `createList` | Creates a new list on Google Maps and adds the specified places to it. 
 **@param listName**: The name of the list to create (e.g., "Anchorage 2025"). 
 **@param places**: A list of place names to add to the list. 
 **@return** true if the list was created successfully and all places were added, false otherwise. |
| | `deleteList` | Deletes saved lists from Google Maps. 
 **@param listName**: All lists whose names contain listName will be deleted. 
 **@return** true if no matching list was found or all such lists were deleted successfully, false otherwise. |

