# OpenReview forum: "Position: Web Agents Should Use Typed Actions Instead of Click-Based Browsing"
_ICML.cc/2026/Position_Paper_Track — ICML 2026 Position Paper Track regular_

### Official Review · Reviewer_Xur5 · 2026-03-12

**Significance:** 3
**Argument Clarity:** 2
**Rating:** 4
**Confidence:** 4

**Questions:**

See the strengths and weaknesses part.

**Alternative Views Section:**

Yes

**Compliance With Llm Reviewing Policy A Conservative:**

Affirmed.

**Discussion Potential:**

2

**Paper Summary:**

This paper argues that Web Agents are currently brittle and inefficient because they rely on low-level actions like clicks and keystrokes. To solve this, the authors propose a semantic layer of Web Verbs. These verbs are high-level, typed functions that package site capabilities into stable, programmable commands. Using Web Verbs allows agents to execute concise, reliable, and auditable workflows instead of fragile step-by-step UI traces.

**Position:**

Yes

**Position In Title:**

Yes

**Related Work:**

3

**Strengths And Weaknesses:**

Strength
===

1. By moving away from the traditional perception-action loop that requires constant re-evaluations of the page state, the proposed method can reduce the number of inference cycles. This makes the execution pipeline much more efficient and scalable.
2. The reliance on low-level primitives makes current agents incredibly brittle, as minor UI updates can break long-horizon behaviors. Abstracting interactions into typed actions effectively decouples the agent's intent from surface-level UI features, directly mitigating low-level execution errors and improving reliability.
3. By elevating the interaction layer, the model is freed from managing coordinates and layout elements. Instead, the agent can leverage its core reasoning strengths to synthesize structured programs, manage explicit control/data flows, and handle multi-step planning. This shift strongly aligns with the natural strengths of modern large language models.

Weakness
===

1. The proposed architecture relies heavily on web developers to provide and define these actions. However, there is no guarantee that different developers will abstract their site's capabilities at the same semantic level. If one website exposes highly granular verbs while another provides only macro-level verbs, this inconsistency could severely hinder an agent's ability to generalize and seamlessly compose cross-site workflows.
2. Because the power to define actions rests with individual site maintainers, there is a high risk of ecosystem chaos. While the authors briefly touch upon standardization, relying on decentralized developers to build a cohesive agentic web could lead to a fragmented landscape filled with overlapping, incompatible, or poorly documented verbs, undermining the proposed unification.
3. Abstracting tasks into statically synthesized programmatic scripts is powerful, but it may reduce the agent's ability to react dynamically. The real-world web is messy, featuring unexpected pop-ups, network timeouts, and A/B testing variations. A step-by-step agent perceives and reacts to these dynamic changes on the fly. It is unclear how a pre-compiled program of verbs handles unexpected environmental shifts without defaulting back to the very perception-action loops the authors are trying to replace.

**Support:**

2

---

> ### Author Rebuttal · Authors · 2026-03-31
>
> We thank the reviewer for the thoughtful and supportive comments. We also appreciate the reviewer’s concerns about standardization, cross-site consistency, and handling dynamic web environments.
>
> **(W1) Verb granularity as a design issue for workflow composition.**
>
> We thank the reviewer for raising this issue. The semantic granularity is a key design question for Web verbs, because it directly affects how well agents can compose workflows.
>
> This issue arises first within a single site. How a developer decomposes site functionality into verbs directly shapes how composable the site is for downstream tasks. For example, on a shopping site, a developer could expose search, add-to-cart, and checkout as three separate verbs, or collapse them into one coarse verb that performs the full purchase flow. The former gives the agent much more flexibility. If the user’s task is to compare similar products and buy the cheaper one, that task becomes much harder, or even impossible, if the whole flow is packaged as one coarse action. In this sense, verb design directly affects which user tasks the agent can solve. The same point extends to cross-site workflows. But if each site exposes its in-site subtasks in a sufficiently composable way, then a cross-site task can **be solved by combining those subtasks across sites**.
>
> This places a real burden on developers. Designing good verbs requires thinking carefully about which user tasks should be supported, how to break them into reusable subtasks, and what level of abstraction is most useful when exposing those subtasks. This is not trivial, and we should make that more explicit in the paper.
>
> However, we do not think this makes the direction infeasible. Web GUI design followed a similar path: early websites were often coarse, inconsistent, and poorly structured, but over time the community developed much better design practices through repeated iteration. We expect the same to happen for verb design. In our view, the verb layer is a necessary step toward an agentic web, and one goal of this position paper is to call on the community to develop the standards, tooling, and best practices that can make such interfaces useful in practice.
>
> **(W2) Fragmentation risk and the need for standardization.**
>
> We thank the reviewer for raising this concern. Without shared conventions, decentralized verb design could lead to overlapping, incompatible, or poorly documented verbs. However, we do not see this as a reason against the verb layer itself. The web is **already highly fragmented** at the level of page structure, site scripts, and ad hoc APIs. What Web verbs add is an explicit action layer that can be documented, compared, and standardized. Our claim is not that decentralized developers will automatically produce unification, but that unification becomes **much more possible once the interface is explicit**.
>
> This is why standardization is a central part of our proposal rather than an afterthought. Naming conventions, shared schemas, registries, and documentation norms are exactly the kinds of mechanisms that can reduce fragmentation over time. In our view, the verb layer is a necessary step toward an agentic web because it creates the interface on which such convergence can happen.
>
> **(W3) How verb-based execution can still react to a messy web.**
>
> We agree that the real-world web is messy. Our goal is not to remove runtime adaptation from the system, but to **avoid forcing the agent to handle every routine step through a low-level perception-action loop**.
>
> A well-designed Web verb can absorb much of this routine variability inside its implementation while still exposing a clean typed interface to the agent. For example, a verb may handle retries, session state, transient pop-ups, or minor UI differences internally, without requiring the agent to replan at every step. If an unexpected situation still cannot be handled inside the verb, the verb can return a structured failure or recovery signal, at which point the agent can replan, choose another verb, or fall back to lower-level interaction if needed.
>
> So the point is not to replace all adaptation with a fixed pre-compiled script. The point is to move the adaptation boundary upward. Routine variability should be handled inside the verb when possible, while harder failures can still trigger runtime replanning. More broadly, many of these dynamic page changes are introduced for human-facing web interaction rather than for agent use. In our view, a mature agentic web should not only support agent adaptation, but also avoid unnecessary dynamic behavior that makes reliable execution harder in the first place. We will revise the paper to make this interaction between verbs, runtime checks, fallback behavior, and agent-friendly web design much clearer.

---

### Official Review · Reviewer_G3kE · 2026-03-13

**Significance:** 3
**Argument Clarity:** 3
**Rating:** 4
**Confidence:** 4

**Questions:**

1. Can you discuss the relationship and different between the proposed typed actions and API-based web interaction?
2. Can you show some more empirical results on the strengths of typed actions compared non-typed ones, both with proper tool definitions for the LLM agents?

**Alternative Views Section:**

Yes

**Compliance With Llm Reviewing Policy A Conservative:**

Affirmed.

**Discussion Potential:**

2

**Final Justification:**

The rebuttal has addressed my main concerns.

**Paper Summary:**

This paper advocates that web agents should not rely on click-based web browsing and should use typed actions instead. It argues that merely based on model enhancement alone is not enough for reliable web agents and the web need to expose a semantic layer of typed actions that agent can invoke and compose. With a comprehensive review on related literature, this paper first analyzes the interaction bottleneck of web agents in three aspects (1) the reliability becomes brittle under low-level primitives; (2) API-only interactions failed to support end-to-end user tasks; (3) step-by-step interaction is costly and scales poorly. Then it introduces a verb abstraction of agent web as a concrete instantiation of the proposed semantic layer by (1) defining the requirements such as semantic clarity, typed parameters, structured results, composability, auditability; (2) defining the programmatic compositions of the web verbs. In addition, this paper provides a case study on travel planning and furniture shopping and show that how the web verbs can be leveraged by agents to write programs and produce more reliable results more efficiently.

**Position:**

Yes

**Position In Title:**

Yes

**Related Work:**

3

**Strengths And Weaknesses:**

Strengths
- The position of this paper is well motivated as to propose adding a semantic layer for web agents to improve reliability and efficiency.
- The position is supported by literature and a case study.

Weakness
- Many existing work discussing that API-based web interaction is better than merely with browser interactions [1,2]. This paper, even though define it as typed actions, seems to show a significant difference than the API-based paradigm.
- The case study does not demonstrate too much advantage of the "typed" actions compared to "untyped" ones, to me, it seems to be more of the advantage of writing code and calling APIs.

[1] Song, Yueqi, et al. "Beyond browsing: Api-based web agents." Findings of the Association for Computational Linguistics: ACL 2025. 2025.
[2] Lù, Xing Han, et al. "Build the web for agents, not agents for the web." arXiv preprint arXiv:2506.10953 (2025).

**Support:**

3

---

> ### Author Rebuttal · Authors · 2026-03-31
>
> We thank the reviewer for raising these thoughtful concerns. These are important points that we will clarify more clearly in the paper.
>
> **(W1, Q1) Relationship between typed actions and API-based web interaction.**
>
> We thank the reviewer for raising this connection to prior work on API-based web agents. We will cite these works in the revision.
>
> The relationship between typed actions and API-based web interaction is that typed actions can naturally build on APIs. A verb may be implemented by one API call, several API calls, or a combination of APIs and browser-side workflows. In that sense, our proposal is not against API-based interaction, but to build a higher-level interface **on top of both APIs and browser interactions for agents**, since in our view both are still lower-level substrates for task execution.
>
> The difference can be summarized in three points:
>
> First, as discussed in our response to `Reviewer TRw6 (W1)`, Web verbs are not limited to the API-only boundary. Many important workflows are not available as one clean public API operation, either because sites do not expose a full public API for the task or because the task depends on **browser session state and interactive confirmation**. A Web verb can still expose such a task as one agent-facing action by combining APIs when available and browser-side workflows when needed.
>
> Second, Web verbs sit at a higher level of abstraction than APIs alone. API-based interaction usually exposes service-level operations, while a verb is meant to expose a task-level operation as one reusable action. This reduces the burden on the agent to discover, order, and connect many lower-level steps on its own. It also **lowers repeated execution cost**, since the task-level logic is packaged once and can then be invoked repeatedly rather than reconstructed by each agent run.
>
> Third, Web verbs are typed task-level actions with clear semantics, typed inputs, and structured outputs. This makes them easier for the agent to compose into short programs and easier to inspect and verify. It also **reduces context burden** much more directly: instead of keeping long sequences of low-level actions or loosely connected API calls in context, the agent can operate over a much shorter representation built from typed actions.
>
> **(W2) Why the case studies may look like “just writing code and calling APIs.”**
>
> We thank the reviewer for raising this point. Our point is that typed actions are important because they are what make code synthesis a practical execution model for the agent. If the agent is expected to solve the task by synthesizing a short program, it must know what operations are available, what inputs they take, and what outputs they return. Typed inputs and structured outputs make this explicit. Without that structure, the agent still has to infer how to call the tool, and how to interpret the return value, which makes code synthesis much less reliable.
>
> This is also where the benefit of code-based execution comes from. Once actions are exposed in typed form, the agent can compose them into short procedures with **explicit control flow and data flow**, rather than keeping long low-level interaction traces in context. This makes the resulting workflow **easier to inspect and reuse**, and **less demanding on context length**. So the intended contrast is not simply “code versus no code,” but “typed actions that support program synthesis versus lower-level operations that still leave the composition burden to the agent.” We will revise the paper to make this point much clearer.
>
> **(Q2) More empirical results comparing typed and non-typed actions.**
>
> We refer the reviewer to our response to `Reviewer JnZA (W3)`. In brief, the two examples shown in the submission are only part of our broader testing. Beyond the two examples shown in the paper, we constructed corresponding verbs for **more than 10 websites and created at least 100 distinct complex tasks** to test feasibility across a much wider range of settings. Across these tasks, verb-based execution was **consistently successful**, which supports the view that the approach is practical beyond the two representative cases included in the paper.
>
> We will revise the paper to better state that these two cases are illustrative examples drawn from a larger suite of testing. We would also be glad to release additional cases and artifacts to further demonstrate the feasibility of the approach and support broader community evaluation.

---

> > ### Author Rebuttal · Reviewer_G3kE · 2026-04-02
> >
> > The rebuttal has generally addressed my concerns.

---

### Official Review · Reviewer_JnZA · 2026-03-13

**Significance:** 2
**Argument Clarity:** 3
**Rating:** 4
**Confidence:** 3

**Questions:**

I stated my questions in the review comments. I will discuss with the authors and pose additional questions during rebuttal.

**Alternative Views Section:**

Yes

**Compliance With Llm Reviewing Policy A Conservative:**

Affirmed.

**Discussion Potential:**

2

**Final Justification:**

After reading the author rebuttal, I think some of my concerns have been resolved. Therefore I updated my rating.

**Paper Summary:**

The paper argues that current web agents, which primarily rely on low-level GUI interaction primitives / actions, face a structural ceiling regarding reliability, efficiency, and auditability. To overcome these limitations, the authors propose shifting to a semantic layer composed of Web Verbs, semantically documented functions that expose website capabilities. These verbs can be backed either by traditional server APIs or by wrapping client-side workflows using browser automation tools like Playwright. The authors contend this allows agents to generate explicit programmatic workflows (using control structures like loops and conditionals) rather than executing long, brittle perception-action traces.

**Position:**

Yes

**Position In Title:**

Yes

**Related Work:**

2

**Strengths And Weaknesses:**

## Strength

1. The paper excellently diagnoses a critical flaw in current Web Agent research: the compounding error rate and high latency inherent in step-by-step, GUI-based perception-action loops
2. The analogy comparing typed web actions to high-level programming languages (abstracting away "assembly-level" clicks) is pedagogically highly effective and clarifies the proposed paradigm shift.

## Weakness

1. The authors suggest that if a site lacks an API, developers can implement a verb by "programmatically driving the browser using an automation library such as Playwright". However, Playwright scripts are brittle precisely because they depend on DOM or a11ytree. If the burden of maintaining these scripts falls on the website owners, why wouldn't they simply expose a REST API, which is far more stable and cheaper to maintain? If the burden falls on third-party developers, the brittleness hasn't been solved; it has merely been shifted from the LLM's inference time to the human developer's maintenance backlog.
2. The core proposal wrapping multi-step GUI interactions into reusable, typed functions using tools like Playwright is very similar to typical robotic process automation and traditional screen-scraping macros. The paper repackages this practice as "Web Verbs" without adequately acknowledging the history or explaining how this conceptually differs from legacy methods, aside from the fact that an LLM is the entity calling the script.
3. The paper relies on two highly simplified "toy" case studies (Travel Planning and Furniture Shopping) to demonstrate the superiority of programmatic composition. These isolated examples are anecdotal and insufficient to prove that this approach scales reliably across the diverse and adversarial reality of the open web

**Support:**

2

---

> ### Author Rebuttal · Authors · 2026-03-31
>
> We thank the reviewer for raising these thoughtful concerns. These are important points that we will clarify more clearly in the paper.
>
> **(W1) Why verbs are still useful when Playwright scripts can be brittle.**
>
> We agree with the reviewer that raw Playwright scripts can be brittle when they rely on unstable DOM structure or accessibility trees. Our claim is **not that today’s browser automation stack is already a complete answer**. Rather, this is exactly why our Standardization section argues that the web should expose **more stable agent-facing anchors**, such as persistent element-level attributes that do not change across routine page updates. In other words, our position is not that current browser automation is already ideal, but that a typed action layer and a more agent-ready web substrate should move forward together.
> At the same time, simply exposing a REST API does not fully resolve the problem. Many important workflows are not available as one clean public API operation. They may depend on browser-session state, interactive confirmation, or other steps that are tied to the live web session. In such cases, as discussed in our response to `Reviewer TRw6 (W1)`, a browser-backed verb gives a site another way to expose the operation as one typed action, without requiring the whole workflow to first be turned into web APIs.
>
> We also agree that verbs move some work from online inference to offline engineering, but we view this as **a main systems advantage** rather than a flaw. For popular sites, many agents repeatedly attempt very similar workflows. Without verbs, each agent pays that planning cost at runtime and often reconstructs nearly the same interaction sequence on its own. With a verb, that effort is done once and then reused many times. This **reduces repeated planning cost, makes behavior more consistent, and puts maintenance in one place**.
>
> More broadly, browser-backed verbs are not meant to be the final form of the web, nor are they meant to replace good APIs where those APIs already exist. They are a practical path for sites whose important workflows are not yet exposed through stable public interfaces. In this sense, verbs can serve as a transitional design while the web moves toward better support for agent-facing interfaces.
>
> **(W2) How Web verbs differ from RPA, macros, and screen-scraping.**
>
> We thank the reviewer for raising this connection. We agree that Web verbs are related to earlier forms of automation such as robotic process automation, browser macros, and screen-scraping scripts. We do not claim novelty in the basic idea of wrapping a multi-step workflow into reusable code, and we should acknowledge this history more clearly in the paper.
>
> The key point of Web verbs is different. Our position is not simply to automate a workflow with a script, but to expose that workflow to the agent as a typed task-level action with clear meaning, inputs, and outputs. In this view, the script is only one possible implementation detail. What matters is the interface given to the agent.
>
> This differs from many legacy automation setups because Web verbs are meant to serve as **reusable building blocks** that an agent can choose, fill in, and combine at runtime, rather than fixed workflow scripts. They are also meant to form a shared semantic layer for composition, inspection, and logging. So the distinction is not just that an LLM is calling the script, but that the wrapped workflow is presented as an agent-facing interface. We will revise the paper to make this lineage clearer and to sharpen the distinction between workflow scripts as implementation artifacts and Web verbs as typed interfaces for agents.
>
> **(W3) Why the two case studies are representative rather than toy examples.**
>
> We would like to clarify that the two case studies in the paper are not toy examples. They were selected under space constraints because they show the kinds of task-level composition problems that our paper is about. In both cases, success requires more than a short sequence of local actions: the agent must keep track of intermediate results and perform a computation over the task as a whole.
>
> The two examples shown in the submission are only part of our broader testing. Beyond the two examples shown in the paper, we **constructed corresponding verbs for more than 10 websites and created at least 100 distinct complex tasks** to test feasibility across a much wider range of settings. Across these tasks, verb-based execution was **consistently successful**, which supports the view that the approach is practical beyond the two representative cases included in the paper.
>
> We will revise the paper to better state that these two cases are illustrative examples drawn from a larger suite of testing. We would also be glad to release additional cases and artifacts to further demonstrate the feasibility of the approach and support broader community evaluation.

---

> > ### Author Rebuttal · Reviewer_JnZA · 2026-04-04
> >
> > thanks for the rebuttal, I updated my rating.

---

### Official Review · Reviewer_TRw6 · 2026-03-17

**Significance:** 2
**Argument Clarity:** 3
**Rating:** 4
**Confidence:** 3

**Questions:**

1) To what extent will the current problems with Web APIs be shared by Web verbs? If you feel that they would not be shared, why not?

**Alternative Views Section:**

Yes

**Compliance With Llm Reviewing Policy A Conservative:**

Affirmed.

**Discussion Potential:**

2

**Final Justification:**

In our final rebuttal interaction, the authors addressed several of my concerns by successfully differentiating between the scenarios by which Web Verbs and Web APIs are developed. I’m still concerned that these differences were not clearer in the original paper, but that can (and should) be fixed in future versions of the paper. The idea that Web Verbs could be produced by third-party developers should be made much clearer. I’ve raised my score to a borderline accept and lowered my confidence.

**Paper Summary:**

The authors advocate for what they term “Web verbs” as a way of facilitating the use of the Web by intelligent agents. The authors contrast Web verbs with prevailing approaches: click-based interaction with GUIs and interaction with Web APIs.

**Position:**

Yes

**Position In Title:**

Yes

**Related Work:**

3

**Strengths And Weaknesses:**

The paper’s critique of the difficulties of agent-based interaction with Web GUIs is strong. Clearly, it is very challenging to create agents that interact with interfaces designed for non-expert humans. Creating successful agents would be much less difficult and far simpler if most websites had a Web verbs interface.

However, the paper is less persuasive when it critiques Web APIs. Indeed, the proposal for Web verbs seems largely a call for better and more widely available Web APIs. Some Web developers have long maintained that online systems are best built by first creating an API, and then developing websites, apps, and other tools that interact with that API layer. Many (though still a small minority) of websites have such APIs. The authors offer at least two substantial critiques of current Web APIs. First, they note that public APIs are relatively rare: “API-only agents are limited not because APIs are weak, but because most sites do not expose a stable, public API surface for completing end-to-end user workflows.” Second, they note that such APIs often have explicit limitations in terms of free and unlimited public access:  “Access may require API keys, OAuth app registration, or strict quotas. Moreover, many consequential workflows depend on browser-session state and interactive confirmations, such as re-authentication, policy acknowledgements, and email or SMS verification.” These problems identified with real-world Web APIs would seem to be likely to apply to any real-world implementation of Web verbs. Given this, the authors’ proposals should detail how to overcome these limitations, but they do not.

A minor point: At the beginning of Section 2, Sutton & Barto’s seminal text on reinforcement learning seems an odd reference for the perception-action loop of most Web-based agents.

**Support:**

2

---

> ### Author Rebuttal · Authors · 2026-03-31
>
> We thank the reviewer for the careful reading and for raising an important concern about the relationship between Web APIs and the Web verbs proposed in our paper.
>
> **(W1) How Web verbs can overcome the limitations of Web APIs.**
>
> We agree with the reviewer that this point should have been made clearer. Our claim is **not that Web verbs remove every practical constraint that affects real-world APIs**. Instead, we view the verb layer as **a necessary practical step toward an agentic web**, even though exposing stable Web verbs for meaningful workflows still takes developer effort and wide adoption will still need community support and standardization.
>
> However, our proposal is not simply a call for better or more widely available public APIs:
>
> **(W1.1) Why the rarity problem does not generally arise for Web verbs.**
>
> A key limitation of Web APIs is that many sites do not expose a stable public API surface for complete user workflows. The reviewer notes that some developers have long preferred to build systems by first creating an API and then building websites, apps, and other tools on top of that API layer. We agree that this can be a good software engineering practice. However, it does not follow that a site will expose a stable public API for the user workflow, or that such an API is the right agent-facing interface for the task.
>
> Exposing a public API usually means packaging functionality as a stable backend product for outside developers, with its own access model, documentation, versioning, and support burden. Many sites do not do this for user workflows. A Web verb has a different and lighter requirement: the site only needs to expose an **agent-facing typed action for a task**, backed by APIs when available or by a maintained browser-side workflow when not.
>
> For this reason, the rarity problem does not transfer in the same form. Web verbs may still depend on site adoption, but they are not limited by the narrower condition that a site must already offer a public API for the workflow. In this sense, our proposal is not simply a call for more public APIs, but for a different agent-facing interface for the web.
>
> **(W1.2) Support for browser-state workflows.**
>
> A second limitation discussed in the paper is that many important workflows are not just missing from public APIs, but are **tied to browser-session state and interactive confirmation**. In practice, many important actions require the user to already be signed in, confirm the action step by step, or complete verification. These are a common part of how real sites handle security, consent, and session continuity. In such cases, the task is often bound to the live web session rather than exposed as a clean external API operation. As a result, even when internal service calls exist, an outside agent often cannot complete the full task through public API access alone.
>
> This is another reason why the limitations of current Web APIs do not apply to Web verbs in the same way. A Web verb can still expose such a task as one typed action by wrapping a maintained browser-side workflow, for example through Playwright-based browser automation. For example, changing the email address of an account on a live website often requires the user to already be signed in, enter the password again, receive a verification code, and confirm the change in that same session. A browser-side Web verb can package this session-bound process as one task-level action, even when no single public API supports it end to end.
>
> **(Q1) To what extent are the current problems with Web APIs shared by Web verbs.**
>
> Some limitations are shared. In particular, exposing a stable interface for meaningful workflows still takes developer effort, and wide adoption will likely need community support and standardization. We should state this more clearly in the paper. This is also why we present the paper as a position paper: we see the verb layer as one step toward an agentic web and hope to push the community in this direction.
>
> However, as discussed in W1, the main limitations of current Web APIs **do not apply** to Web verbs in the same way. In particular, Web verbs are not tied to the same notion of public API availability, and they can also support tasks that depend on browser session state and interactive confirmation.
>
> More generally, Web verbs are a higher-level abstraction than Web APIs from the agent’s point of view. An API often exposes service-level operations, while a verb is meant to expose a task-level action with clear semantics, typed inputs, and structured outputs. This reduces the burden on the agent to discover, combine, and manage many low-level steps on its own, and makes the resulting workflow **easier to reuse, inspect, and verify**.
>
> **Minor point on the citation in Section 2.**
>
> We thank the reviewer for pointing out the citation at the beginning of Section 2. We will check if the citations in the paper are appropriate.

---

> > ### Author Rebuttal · Reviewer_TRw6 · 2026-04-05
> >
> > Thanks to the authors for clarifying their position regarding my concerns about the practicality of the Web verbs proposal, given the existing (substantial) experience with Web APIs. The authors argue that Web Verbs are substantially different from Web APIs. For example, they state that: "...Web verbs are a higher-level abstraction than Web APIs from the agent’s point of view."
> >
> > This doesn't really address the point that I attempted to make in my original review. I'm not particularly concerned about how things look from the *agent's* point of view. My concern is about the *developer's* and *manager's* point of view. That is: Some set of forces has resulted in Web APIs being relatively rare, and I'm concerned that the authors have not understood this set of forces and that those forces will apply equally (or even more strongly) to Web Verbs. Examples of such forces might include: (1) *Developer effort* (developers are too busy deploying and maintaining a human accessible website to consider also supporting a Web API); (2) *Financial incentives* (there are currently few ways to monetize agentic access to a website, given that agents don't read and click on ads, so there are strong financial disincentives to providing a Web API); and (3) *Collection of human feedback* (the business model of some websites is to collect data on human preferences, so they are less interested in serving automated agents). Note that I'm not saying that these forces *are* responsible for the relative lack of Web APIs, but they seem likely and they would equally limit the utility of Web Verbs. That's the point of a major part of my review, and it's not addressed in the authors' reply.

---

### Decision · Program_Chairs · 2026-04-30

**Decision:**

Accept (regular)

**Comment:**

This paper presents a well-argued and timely position on the structural limitations of current web agents, proposing a shift toward a semantic layer of typed Web Verbs. Initially, the review panel raised several valid critiques, including differentiation from Web APIs, brittleness and legacy automation, and ecosystem fragmentation. During the rebuttal, the authors effectively addressed these concerns. They clarified that Web Verbs are a higher-level, agent-facing abstraction that can securely wrap both public APIs and live browser-session states (which APIs often fail to support). They successfully differentiated Verbs from legacy macros by emphasizing their typed, compositional nature, which allows LLMs to synthesize programmatic workflows rather than merely executing rigid scripts. All reviewers ultimately recognized the high relevance and pedagogical clarity of the proposed paradigm shift. The paper serves as a call to action for standardization in the agentic web and is a highly appropriate fit for the Position Paper track.